# ROBUST SELF-SUPERVISED LEARNING IN HETEROGENEOUS GRAPH BASED ON FEATURE-TOPOLOGY BALANCING

## ABSTRACT

In recent years, graph neural network (GNN) based self-supervised learning in heterogeneous information networks (HINs) has gathered considerable attention. Most of the past studies followed a message passing approach where the features of a central node are updated based on the features of its neighboring nodes. Since these methods depend on informative graph topology and node features, their performance significantly deteriorates when there is an issue in one factor. Moreover, since real-world HINs are highly noisy and validating the importance of attributes is challenging, it is rare to find cases where both the graph topology and node features are of good quality. To address this problem, we make the first model that can explicitly separate the graph topology and features in the heterogeneous graph by proposing the novel framework BFTNet (robust self-supervised heterogeneous graph learning using the **B**alance between node **F**eatures and graph **T**opology). BFTNet employs a knowledge graph embedding module focusing on global graph topology and a contrastive learning module dedicated to learning node features. Thanks to the novel structure that handles graph topology and node features separately, BFTNet can assign higher importance to one factor, thereby allowing it to effectively respond to skewed datasets in real-world situations. Moreover, BFTNet can improve performance by designing the optimal module suited for learning the topology and features, without sacrificing the performance of one modality to reflect the characteristics of the other modality. Lastly, BFTNet implemented a novel graph conversion scheme and representation fusion method to ensure that the representation of topology and features are effectively learned and integrated. The self-supervised learning performance of BFTNet is verified by extensive experiments on four real-world benchmark datasets, and the robustness of BFTNet is demonstrated with the experiments on noisy datasets. The source code of BFTNet will be available in the final version.

## 1 INTRODUCTION

Recently, for its versatility to represent real-world where diverse entities interact in various ways, heterogeneous information networks (HINs) have received significant attention in various fields such as social media analysis Prangnawarat et al. (2015), recommendation system Fan et al. (2019), and biomedical databases Wang et al. (2021c). In line with this trend, research on how to effectively utilize HINs for downstream tasks such as link prediction Fu et al. (2020), community detection Luo et al. (2021), and knowledge graph construction Zeng et al. (2018) has been actively conducted. In particular, finding low-dimensional vector representation that effectively captures the information of complex HINs has been revealed to be useful for various downstream tasks Tang et al. (2015).

From the early stages of representation learning in HINs, message passing based methods have been a major approach Wang et al. (2021a); Ren et al. (2020); Jing et al. (2021). However, previous methods fail to generalize on various graph datasets. One of the primary reasons for this problem is that the assumption underlying previous studies, which suppose that both graph topology and node features are reliable, is frequently violated in real-world graphs Zhao et al. (2021).

(a) message passing based method          (b) Score function based method

Figure 1: Difference between graph topology learning methods based on (a) message passing and (b) score function in a graph with missing edges. The symbol $f(\cdot, \cdot, \cdot)$ indicates the score function. When one edge is missing, the representation of node $A$ can change significantly in method (a), while that in method (b) is robust against edge missing.

The one scenario that violates the aforementioned assumption is when the initial node features of HINs contain a lot of unnecessary information. For example, due to the difficulties in the data curation process, many social networks have irrelevant or redundant node attributes Rostami & Oussalah (2022). In such cases, the model should prioritize learning from the overall topology of the graph over the features of individual nodes. However, for methodologies based on message passing Wang et al. (2021b); Yang et al. (2022); Park et al. (2020), it is impossible to prioritize one aspect over the other. Moreover, due to the over-smoothing problem Li et al. (2018) of GNN, expanding the receptive field by stacking more GNN layers is highly challenging. To sum up, not only are message passing methods unable to prioritize learning the global graph structure, but they also have limitations in their ability to learn the graph topology.

On the contrary, there are situations where the graph topology does not hold reliable information about node representation. For instance, obtaining a reliable graph structure in recommendation systems is challenging because privacy concerns and erroneous information such as mis-clicks is rampant Zhao et al. (2021). However, in many SSL methods for HINs, the training objective often entirely relies on the structural features of the graph Wang et al. (2021b); Yang et al. (2022). In such cases, the training objective of the SSL methods can be largely distorted even with minor edge perturbations, showing poor robustness.

To address this problem, we propose an exceptional SSL method, BFTNet (robust self-supervised heterogeneous graph learning using the **B**alance between graph **T**opology and node **F**eatures). BFT-Net consists of two main modules: the knowledge graph embedding module and the contrastive learning module. The former module specializes in learning the graph topology, and the latter focuses on individual node features. The advantages of BTFNet can be explained as follows:

Due to the methodology of separating the graph topology and node feature, which has not been attempted in previous studies, BFTNet is able to prioritize a certain factor and does not overly rely on single aspects of the graph. The weights of graph topology and node features are adjustable based on their relative importance in BFTNet. Therefore, BFTNet has the flexibility to adjust learning at an optimal balance point, even in significantly skewed datasets. Moreover, BFTNet can remain stable in the graph with missing edges. For the message passing based method in Figure 1 (a), the representation of node **A** is determined by summation of the messages from adjacent nodes **B**, **C**, and **D**. Thus, if the edge between **A** and **D** is missing, the representation of node **A** can change significantly. On the other hand, for the method using score function based depicted in Figure 1 (b), the representation of **A** is determined by the score function $f(\cdot, \cdot, \cdot)$ between **A** and its adjacent nodes. Therefore, even if the edge between **A**–**D** is missing, the relative position between **A**–**B**, and **A**–**C** are trained to be maintained, resulting in robustness against missing edges.

BTFNet can also utilize the optimal learning architecture tailored to the graph topology and node features. For example, BFTNet outperforms traditional methods in terms of learning the graph topology. As mentioned above, an edge and the two nodes connected by the edge in BFTNet are trained to conform to a certain rule. These rules possess a composite property, allowing for easy combinations of edges to represent long distances Abboud et al. (2020); Sun et al. (2019). Therefore, in BFTNet, even if two nodes are relatively far apart, the representation of the nodes is learned to

satisfy the rule defined by composite edges, enabling better learning of the position where particular nodes exist within the overall graph topology.

Furthermore, we proposed a novel graph conversion method and representation fusion method so that BTFNet can focus on target node-relevant triples and integrate representations from two modalities. Such improvement is verified with experiments with baseline models and the ablation study. Accordingly, our main contribution can be described as follows:

- We propose a novel SSL method, BFTNet. BFTNet is the first method capable of explicitly reflecting the relative importance of graph topology and node features.

- We propose a highly effective method by formulating the optimal SSL methods for each modality of the graph and implementing a graph conversion scheme. The method is validated by showing the best performance in four benchmark datasets.

- We propose a highly robust method by devising a model that can adequately learn the node features and graph topology through a mutually upholding structure. The utility of the model is verified through experiments with noisy data.

## 2 PRELIMINARIES

**Heterogeneous Information Networks** HIN is defined as a graph $\mathcal{G} = (\mathcal{V}, \mathcal{A}, \mathcal{X}, \mathcal{E}, \mathcal{R})$, where $\mathcal{V}$, $\mathcal{A}$, $\mathcal{X}$, $\mathcal{E}$, and $\mathcal{R}$ denote a set of nodes, node types, node features, edges and edge types, respectively, which satisfies $|\mathcal{A} + \mathcal{R}| > 2$. $\boldsymbol{v}_i \in \mathcal{V}$ represents the $i$-th node element of $\mathcal{V}$, with its node feature denoted as $\boldsymbol{x}_i \in \mathcal{X}$. $\boldsymbol{a}_i \in \mathcal{A}$ and $\boldsymbol{r}_j \in \mathcal{R}$ represent the $i$-th node type element of $\mathcal{A}$ and the $j$-th edge type element of $\mathcal{R}$, respectively.

**Knoweledge graph** The knowledge graph can be defined as a set of triples $\mathcal{D} = \{(h, r, t)\}$, where $h, t \in \mathcal{V}$ represent head node and tail node, and $r \in \mathcal{R}$, represents the edge connecting $h$ and $t$.

Because of the similarity of the definition of HINs and knowledge graphs, knowledge graphs can be regarded as an edge-rich case of HINs Wang et al. (2016).

**Metapath** A metapath $\mathcal{M}$ of a heterogeneous information network $\mathcal{G}$ is defined as a subset of paths in $\mathcal{G}$, which are in a format where node types and edge types are intertwined in an alternating manner. For example, $\boldsymbol{a}_1 \boldsymbol{r}_1 \boldsymbol{a}_2 \boldsymbol{r}_2 .... \boldsymbol{r}_l \boldsymbol{a}_{l+1}$ metapath describes a set of paths within the graph $\mathcal{G}$ where nodes and edges are connected in the specified order of types $\boldsymbol{a}_1 \boldsymbol{r}_1 \boldsymbol{a}_2 \boldsymbol{r}_2 .... \boldsymbol{r}_l \boldsymbol{a}_{l+1}$.

**Self-supervised learning in HINs** Given a HIN $\mathcal{G} = (\mathcal{V}, \mathcal{A}, \mathcal{X}, \mathcal{E}, \mathcal{R})$, the problem of SSL in HINs is to learn a $d$-dimensional vector representation $\mathbf{h}_i^{\text{target}} \in \mathbb{R}^d$ for target node $v_i^{\text{target}} \in \mathcal{V}$ without using any labels. Note that only a specific node type is a target node type, which is a learning target of SSL for HINs (Blue squares in Figure 1 and 2).

## 3 THE PROPOSED MODEL: BFTNET

In this section, we explain the architecture of BFTNet (Figure 2), a novel and robust SSL-based model that can reflect the relative importance between the node feature and the graph topology.

### 3.1 HIN CONVERSION BASED ON TRANSITION NODE-WISE METAPATH

We propose a novel graph conversion scheme to enhance the learning performance of the graph topology in the Knowledge Graph module. As mentioned in Section 1, BFTNet makes triples follow the rule defined by the score function. Therefore, nodes that are frequently included in triples participate more often in the training, while nodes that are less included participate less. This setting is problematic for SSL in HINs since it only considers the representation of the target node. Therefore, to make the knowledge graph module focus on target nodes, we have devised a new *graph conversion based on transition node-wise metapath* as shown in (#1 in Figure 2). Within this strategy, the original triples are modified such that both ending nodes serve as target nodes, significantly enhancing the graph topology learning performance. The specific graph conversion process starts from finding the transition node-wise metapath in the HIN $\mathcal{G}$.

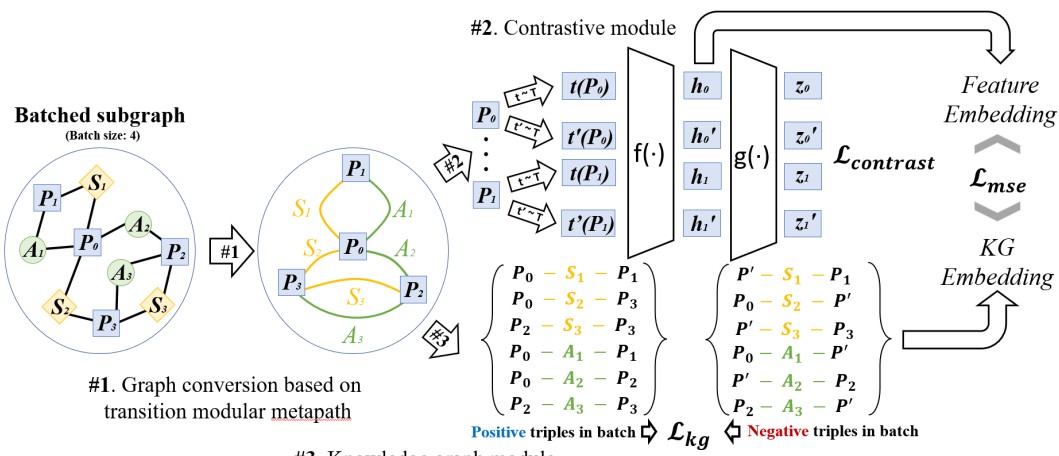

Figure 2: The architecture of BFTNet. This figure is based on ACM dataset with three node types (author(A), paper(P), and subject(S)). The target node type is paper(P). **#1** Given HIN is converted based on transition node-wise metapath, to learn more precise graph topology representation. This is basically the process of converting non-target nodes in the given graph into edges in a converted graph. After graph conversion, BFTNet encodes two node representations; **#2** one is derived from the node features by the contrastive module **#3**, and the other is learned from the graph topology by knowledge graph module. During the learning process, two representations supervise each other ($\mathcal{L}_{mse}$) and learn an integrated (even balanced) node representation.

**Definition 1.** *Transition node-wise metapath* $\mathcal{M}(\boldsymbol{a}_t, \boldsymbol{a}_{nt})$ *for target node type* $\boldsymbol{a}_t \in \mathcal{A}$ *and non-target node type* $\boldsymbol{a}_{nt} \in \mathcal{A}$ *is defined as the set of shortest paths that start from* $\boldsymbol{a}_t$ *type nodes, passing through* $\boldsymbol{a}_{nt}$ *type nodes and arriving different* $\boldsymbol{a}_t$ *type nodes. The length of the transition node-wise metapath is determined as the number of edges in the path. If* $\boldsymbol{r}_k^{t,nt}$ *is the k-th path in* $\mathcal{M}(\boldsymbol{a}_t, \boldsymbol{a}_{nt})$, *two nodes at the ends of the path are defined as a* ***transition node-wise metapath based neighbor*** *of each other. Excluding these two nodes,* $\boldsymbol{a}_{nt}$ *type nodes in the path are defined as* ***transition nodes***.

Based on transition nodes and transition node-wise metapath based neighbors, the graph conversion can be defined as follows.

**Definition 2.** *Graph conversion based on transition node-wise metapath. For all* $\boldsymbol{a}_{nt} \in \mathcal{A}$ *defined in a HIN* $\mathcal{G}$, *if two nodes* $v_i, v_j \in \mathcal{V}$ *are* ***transition node-wise metapath based neighbor*** *connected by* $\boldsymbol{r}_k^{t,nt} \in \mathcal{M}(\boldsymbol{a}_t, \boldsymbol{a}_{nt})$, *add* $v_i$ *and* $v_j$ *to the converted HIN* $\mathcal{G}'$, *and connect them with edges of the edge types corresponding to the transition nodes of* $\boldsymbol{r}_k^{t,nt}$. *If there is a direct connection between two target nodes, add them to the* $\mathcal{G}'$ *and connect them with the same edge type in* $\mathcal{G}$.

For example, in Figure 2, $\mathbf{P}_0\mathbf{A}_2\mathbf{P}_2$ is the shortest metapath using target node type $\mathbf{P}$ and node type $\mathbf{A}$. Therefore, $\mathbf{P}_0\mathbf{A}_2\mathbf{P}_2 \in \mathcal{M}(\mathbf{P}, \mathbf{A})$ and its transition node is $\mathbf{A}_2$. Since $\mathbf{P}_0$ and $\mathbf{P}_2$ is *transition node-wise metapath based neighbor* with edge $\mathbf{A}_2$, converted graph in Figure 2 contains triple $(\mathbf{P}_0, \mathbf{A}_2, \mathbf{P}_2)$. Since an edge type is assigned according to the transition node, the total number of edge types can be at most the number of nodes present in the original HIN $\mathcal{G}$. Thus, even after going through the proposed conversion, it is possible to construct the model using the same number of parameters as used for the original HIN $\mathcal{G}$.

### 3.2 CONTRASTIVE LEARNING MODULE

For node feature learning, we formulate a node-level contrastive learning module inspired by representation learning in vision domain Chen et al. (2020) (#2 in Figure 2). In this module, two views are created for each individual node feature within a batch, and then the module is trained to maximize agreement between them and minimize agreement between those from different nodes. Two views are generated by introducing two different Gaussian noises $\epsilon_i, \epsilon_j$ into the node feature.

$$t_i(\boldsymbol{x}_k) = \boldsymbol{x}_k + \epsilon_i \quad t_j(\boldsymbol{x}_k) = \boldsymbol{x}_k + \epsilon_j \quad t_i, t_j \sim \mathcal{T} \tag{1}$$

where $\boldsymbol{x}_k$ is a initial feature of node k, $t_i$ and $t_j$ denote augmentations in augmentation pool $\mathcal{T}$. The two generated views are transformed into low-dimensional representation $h$ through the feature encoder $f(\cdot)$. Afterward, projection $g(\cdot)$ is employed to transform the $h$ into the latent space representation $z$, which is used in the calculation of the loss function. In this study, two FC layers were used for $f(\cdot)$ and $g(\cdot)$.

$$h_{k,i} = f(t_i(\boldsymbol{x}_k)) \quad z_{k,i} = g(h_{k,i}) \quad h_{k,j} = f(t_j(\boldsymbol{x}_k)) \quad z_{k,j} = g(h_{k,j}) \tag{2}$$

Then, the following contrastive loss function is defined to enhance the consistency between correlated views, in comparison to the views from the other nodes.

$$\mathcal{L}_{\text{contrast}} = \frac{1}{N} \sum_{i=1}^{N} \left[ -\log \left( \frac{\exp(\text{sim}(z_{n,i}, z_{n,j})/\tau)}{\sum_{n'=1, n' \neq n}^{N} \exp(\text{sim}(z_{n,i}, z_{n',j})/\tau)} \right) \right] \tag{3}$$

In the above equation, $N$ is the number of nodes in the batch, $\text{sim}(\cdot, \cdot)$ is a cosine similarity function, $z_{n,j}$ is the latent space representation of $n$-th node with $t_j$ augmentation and $\tau$ is the temperature parameter. Through the use of the specified loss function, we can effectively learn feature representation $h_{feat}$ without the aid of topology.

## 3.3 KNOWLEDGE GRAPH EMBEDDING MODULE

Knowledge graph module effectively learns the graph topology. Because real-world HINs contain a significant number of symmetric edge patterns, BFTNet has introduced a rotation-based score function Sun et al. (2019). The score function of the knowledge graph module is as follows:

$$d_r(h_i, h_j) = \|h_i \circ r - h_j\|_1 \tag{4}$$

In the above equation, $h_i$, $h_j$, and $r$ mean the representation of the $i$-th node, $j$-th node, and an edge between the two nodes defined in the graph resulting from the graph conversion defined in section 3.1, and $\circ$ represent Hadamard product. For the learning of the knowledge graph module, we minimize the score of the positive triples $(h, r, t)$ and maximize the score of the negative triples $(h', r, t')$ (#3 in Figure 2). For each iteration, BFTNet randomly selects $N$ number of target nodes from the converted graph. Subsequently, positive triples $(h, r, t)$ for the iteration are constructed from the triples involving these $N$ nodes. Additionally, within each iteration, half of the positive triples have their head node $h$ changed, and the other half have their tail node $t$ changed, thereby forming $l$ negative triples per one positive triple. Consequently, the loss function for the knowledge graph module is defined as follows.

$$\begin{aligned} \mathcal{L}_{kg} = &- \log \sigma(\gamma - d_r(h, t)) \\ &- \sum_{i=1}^{l} p(h'_i, r, t'_i) \log \sigma(d_r(h'_i, t'_i) - \gamma), \end{aligned} \tag{5}$$

where $\sigma$ is the sigmoid function, $l$ is the number of negative triples per one positive triple, $\gamma$ is the predefined margin, and $p(h'_i, r, t'_i)$ is the weight of negative triple $(h', r, t')$ defined in Sun et al. (2019). Finally, through the knowledge graph module, BFTNet can learn graph topology-based representation $h_{kg}$ for the N nodes within the batch.

## 3.4 OPTIMIZATION

At the last step of each iteration, the mean squared error (MSE) loss between the two representations from two modalities is calculated for their alignment. This loss also facilitates mutual supervision by enabling each module to exchange information.

$$\mathcal{L}_{mse} = MSEloss(h_{feat}, h_{kg}) \tag{6}$$

To balance feature and topology learning and control their merging, we introduce two hyperparameters: the balance hyperparameter $\alpha$ and the alignment hyperparameter $\beta$.

$$\mathcal{L}_{total} = \beta\mathcal{L}_{mse} + (1 - \beta)(\alpha\mathcal{L}_{contrast} + (1 - \alpha)\mathcal{L}_{kg}) \tag{7}$$

The balance hyperparameter $\alpha$ determines the weights of the losses from the two modules, thereby determining the relative importance of the two modalities. Therefore, adjusting $\alpha$ allows the model to accommodate all cases where the dataset is relatively more important in terms of features or topology. Furthermore, the introduction of the alignment parameter $\beta$ determines the extent to which the information derived from the node feature and the graph topology will be aligned. For instance, if the information from either the graph topology or node feature significantly lags behind the other, aligning the two representations will lead to a loss of crucial information. In contrast, in cases where both representations are informative, it would be effective to sufficiently align the two representations by compensating for each other's deficiencies. In conclusion, by introducing the two hyperparameters, BFTNet is enabled to effectively handle a broad range of datasets, overcoming the limitations of traditional methodologies. The final representation obtained from BFTNet, reflecting the optimal balance learned through $\alpha$, is given as follows.

$$h_{total} = \alpha h_{feat} + (1 - \alpha)h_{kg} \tag{8}$$

## 4 EXPERIMENTS

### 4.1 EXPERIMENTAL SETUP

**Datasets** We evaluated BFTNet on four benchmark datasets: **IMDB** Yang et al. (2022), **DBLP** Fu et al. (2020), **ACM** Zhao et al. (2020), **MAG** Yang et al. (2022) These datasets have been used in previous related studies Yang et al. (2022); Wang et al. (2021b); Yang et al. (2021); Park et al. (2020). The details of the datasets are presented in Appendix A.1.

**Baselines** For baseline models, one unsupervised homogeneous model and six Self-supervised heterogeneous models are used. The experimental settings of baselines are presented in Appendix A.2.

- unsupervised homogeneous model: GraphSAGE (abbreviated as SAGE) Hamilton et al. (2018)

- Self-supervised heterogeneous model: Mp2vec (abbreviated as M2V) Jiang et al. (2017), DMGI Park et al. (2020), CKD Wang et al. (2022), SHGP Yang et al. (2022), HDMI Jing et al. (2021), HeCo Wang et al. (2021b)

### 4.2 FEATURE - TOPOLOGY BALANCE OF THE DATASETS

To verify our claim that the amount of useful information in node features and graph topology varies across datasets, we designed an experiment to measure the amount of mutual information between the label and two graph components. We trained a contrastive module with the initial node features and a knowledge graph module with the graph topology. Subsequently, we measured the amount of useful information in two modalities through normalized mutual information (NMI) (Table 1). Surprisingly, the amount of useful information in features and topology appeared very different across the

Table 1: Mutual information between node features and labels in four real-world graph datasets

| dataset | feature | topology |
|---------|---------|----------|
| **IMDB** | $4.16 \pm 0.00$ | $4.85 \pm 0.01$ |
| **ACM** | $37.47 \pm 0.02$ | $38.38 \pm 0.03$ |
| **DBLP** | $12.41 \pm 0.03$ | $71.06 \pm 0.01$ |
| **MAG** | $64.61 \pm 0.00$ | $92.33 \pm 0.01$ |

four datasets. DBLP exhibited much higher NMI in topology than in feature, indicating that the dataset contains more crucial information in the topology than in the features. On the other hand, in the case of ACM, the difference in NMI between feature and topology is marginal. This suggests that, compared to DBLP, ACM is a dataset where significant information is leaned toward features.

## 4.3 OBJECT CLASSIFICATION

To evaluate BFTNet in object classification tasks, we adopted the previous methods for the fair comparison Yang et al. (2022). The pre-trained node embeddings were used to train a linear classifier. The training sets were composed by randomly selecting 4%, 6%, and 8% of target nodes. Macro F1 score and Micro F1 score were used as a classification performance metric, and the source code of DMGI Park et al. (2020) was used for calculation. In Table 2, BFTNet demonstrates outstanding object classification performance on IMDB. As shown in Table 1, IMDB is the hardest dataset with the least correlated node feature. However, BFTNet shows more than 10% of improvements compared to all conventional models. These results demonstrate that models that do not learn graph topology separately have limitations in topology learning. Moreover, BFTNet shows the best results on ACM, DBLP, and MAG. This improvement in performance demonstrates the effectiveness of reflecting the relative importance of features and topology, unlike the conventional message passing based methodologies.

Table 2: Object Classification results (%)

| Datasets | Metrics | Train ratio | SAGE | M2V | DMGI | HDMI | HeCo | CKD | SHGP | BFTNet |
|---|---|---|---|---|---|---|---|---|---|---|
| IMDB | Mac-F1 | 4% | 25.03 | 24.08 | 38.02 | 37.74 | 37.52 | 45.31 | 44.28 | **56.88** |
| | | 6% | 26.78 | 24.50 | 38.35 | 37.88 | 38.69 | 45.67 | 45.23 | **56.61** |
| | | 8% | 27.85 | 25.02 | 39.72 | 38.04 | 39.91 | 47.32 | 47.25 | **56.62** |
| | Mic-F1 | 4% | 51.34 | 56.23 | 52.46 | 54.34 | 57.87 | 59.23 | 57.69 | **65.19** |
| | | 6% | 52.45 | 56.50 | 53.34 | 55.36 | 58.11 | 59.50 | 58.72 | **66.31** |
| | | 8% | 53.01 | 55.95 | 54.23 | 56.23 | 59.46 | 60.85 | 60.50 | **67.57** |
| ACM | Mac-F1 | 4% | 48.50 | 52.23 | 87.84 | 86.20 | 88.15 | 89.40 | 88.53 | **91.25** |
| | | 6% | 56.07 | 57.73 | 87.75 | 87.24 | 88.24 | 89.52 | 88.60 | **91.47** |
| | | 8% | 57.32 | 58.30 | 88.32 | 87.32 | 88.75 | 90.15 | 89.89 | **91.78** |
| | Mic-F1 | 4% | 50.32 | 63.34 | 88.34 | 85.62 | 87.29 | 89.21 | 87.73 | **91.11** |
| | | 6% | 62.80 | 64.21 | 87.56 | 86.34 | 88.60 | 89.47 | 87.40 | **91.24** |
| | | 8% | 61.29 | 64.09 | 88.46 | 87.27 | 88.27 | 89.75 | 89.32 | **91.51** |
| DBLP | Mac-F1 | 4% | 72.48 | 88.34 | 88.65 | 90.23 | 90.82 | 90.52 | 90.23 | **91.91** |
| | | 6% | 74.28 | 88.53 | 88.89 | 90.45 | 90.56 | 90.53 | 90.90 | **92.15** |
| | | 8% | 74.92 | 88.34 | 88.50 | 90.54 | 91.76 | 91.63 | 91.21 | **92.31** |
| | Mic-F1 | 4% | 71.35 | 88.10 | 87.92 | 89.98 | 90.46 | 90.45 | 90.83 | **92.51** |
| | | 6% | 72.19 | 88.46 | 88.23 | 89.45 | 90.10 | 90.34 | 90.60 | **92.40** |
| | | 8% | 74.53 | 89.32 | 88.40 | 89.24 | 91.31 | 91.57 | 91.20 | **92.72** |
| MAG | Mac-F1 | 4% | 88.53 | 89.43 | 94.01 | 94.53 | 94.89 | 95.32 | 97.24 | **98.65** |
| | | 6% | 89.93 | 89.51 | 94.12 | 94.28 | 95.87 | 95.56 | 98.30 | **98.82** |
| | | 8% | 90.30 | 89.29 | 94.23 | 94.88 | 95.99 | 96.32 | 98.27 | **98.87** |
| | Mic-F1 | 4% | 89.01 | 88.34 | 94.21 | 95.01 | 95.23 | 95.25 | 98.13 | **98.53** |
| | | 6% | 89.32 | 88.59 | 94.32 | 95.21 | 95.31 | 95.26 | 98.30 | **98.85** |
| | | 8% | 89.64 | 89.13 | 94.54 | 95.33 | 95.33 | 95.47 | 98.47 | **98.83** |

Table 3: Object clustering results (%)

| | IMDB | | ACM | | DBLP | | MAG | |
|---|---|---|---|---|---|---|---|---|
| | NMI | ARI | NMI | ARI | NMI | ARI | NMI | ARI |
| BFTNet | **7.09** | **5.23** | **67.00** | **70.98** | **77.19** | **81.63** | **92.82** | **94.87** |
| SHGP | 6.53 | 2.98 | 57.46 | 57.34 | 75.34 | 79.12 | 90.34 | 92.54 |
| CKD | 5.56 | 2.92 | 56.34 | 57.41 | 76.34 | 80.01 | 88.45 | 91.23 |
| HeCo | 5.50 | 2.12 | 55.46 | 57.94 | 74.41 | 79.90 | 78.27 | 81.48 |
| HDMI | 4.56 | 3.50 | 53.24 | 48.76 | 72.34 | 78.23 | 78.40 | 80.45 |
| DMGI | 3.38 | 2.76 | 51.23 | 47.65 | 69.44 | 74.35 | 69.87 | 72.53 |
| M2V | 1.23 | 1.11 | 43.22 | 35.46 | 62.44 | 63.44 | 39.67 | 43.72 |
| SAGE | 0.50 | 0.8 | 29.34 | 28.23 | 53.44 | 38.40 | 32.34 | 40.23 |

## 4.4 OBJECT CLUSTERING

For the object clustering task, we also followed the previous studies Yang et al. (2022) for a fair comparison. A k-means clustering model was used to cluster the pre-trained node embeddings, and the metrics were NMI(normalized mutual information) and ARI(adjusted random index). In Table 3, BFTNet consistently reports the best results on all datasets, including more than 10% difference in ACM. In particular, overcoming the low-quality node features of IMDB (Table 1), BFTNet reports almost 10% improvement in NMI and 50% improvement in ARI. This result repeatedly supports our claim that BFTNet is more capable of learning topology because it has a module focused on it. For the topology, even recent GNN-based models trained on MAG show a lower NMI score than that of the knowledge graph module alone (92.33 as shown in Table 1). This shows the GNN-based model has fundamental limitations because they cannot learn the topological features separately.

## 4.5 ROBUSTNESS ON HIGHLY NOISY HINS

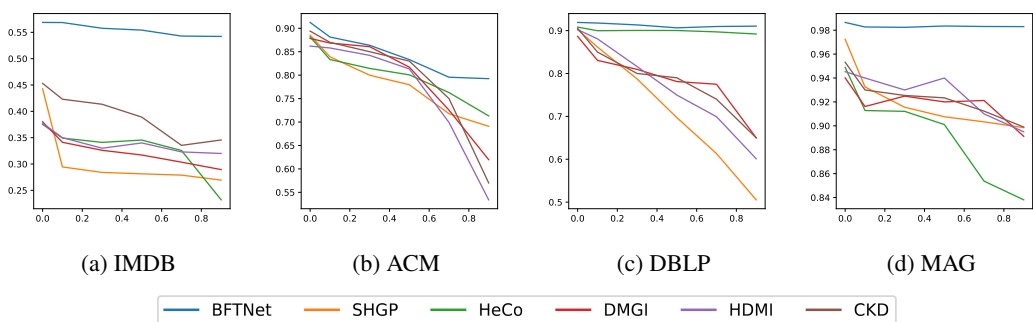

Figure 3: Object classification performance (Y axis, Macro f1 score at 4%) of BFTNet and baseline models under different **feature-masking ratio** (X axis, 0 to 90%).

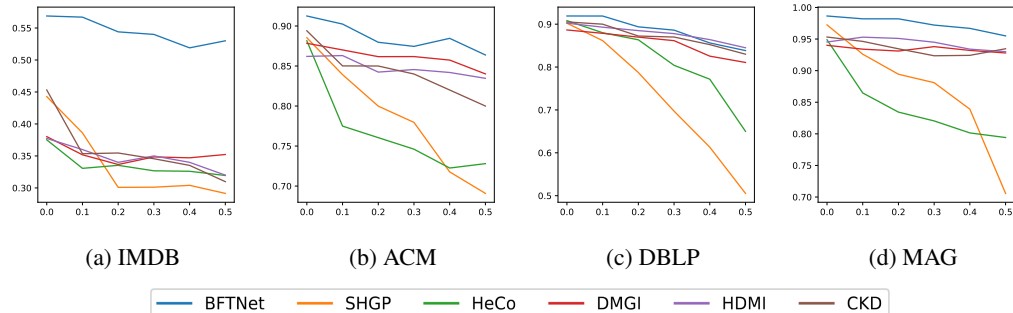

Figure 4: Object classification performance (Y axis, Macro f1 score at 4%) of BFTNet and baseline models under different **edge-dropping ratio** (X axis, 0 to 50%).

On top of the solid performance of BFTNet in HINs with four diverse feature-topology balances, we demonstrated the adaptability of BFTNet in HINs with more extreme conditions. We constrained information from feature and topology by either masking node features or dropping edges. We applied masking based on the dimension most frequently used across all nodes because feature values are zero in most dimensions of the nodes. We applied masking to 10%, 30%, 50%, 70%, and 90% of the entire node features for node feature constraint. Similarly, for graph topology constraint, we dropped 10%, 20%, 30%, 40%, and 50% of the entire edges. Figure 3 and Figure 4 demonstrate the performance of BFTNet in feature-masking scenarios and edge-dropping scenarios, respectively. In Figure 3, despite the increase in noise, BFTNet showed a near-zero performance drop on IMDB, DBLP, and MAG. In ACM, BFTNet showed the best performance compared to the baseline models in all cases. Especially, BFTNet demonstrated a significant performance difference as the noise level became extreme. In the edge-dropping scenario of Figure 4, our model shows exceptional performances in all datasets. BFTNet sustained its performance at the level of SOTA (state-of-the-art) even at high noise levels. These consistent results demonstrate that the model can balance features and topology well across diverse datasets, thereby showcasing its high robustness.

## 4.6 ABLATION STUDY

We further analyzed the contribution of each module and graph conversion to the performance. The result is shown in Table 4. **BFTNet-fx** and **BFTNet-tx** are the models without a contrastive module and a knowledge graph module, respectively. **BFTNet-cx** is the model trained without using the graph conversion. Across all datasets, BFTNet showed better performance than BFTNet-cx. This demonstrates the excellence of our proposed graph conversion. Moreover, BFTNet always performed better than BFTNet-fx and BFTNet-tx, which indicates that the information from each module is effectively merged. Hyperparameter study about hyperparameter $\alpha$ and $\beta$ is presented in Appendix A.3.

Table 4: Ablation study on the object classification task( Macro f1 score at 8%)

|          | IMDB  | ACM   | DBLP  |
|----------|-------|-------|-------|
| BTFNet   | **56.62** | **91.78** | **92.31** |
| BTFNet-cx| 54.20 | 89.28 | 90.21 |
| BTFNet-tx| 36.05 | 85.14 | 69.83 |
| BTFNet-fx| 52.28 | 38.21 | 75.21 |

## 5 RELATED WORK

**Over smoothing problem of Graph Neural Networks** The over-smoothing problem Li et al. (2018), which refers to the inability of GNN to benefit from deep layers, has been identified as a significant issue in GNN. To reduce this problem, regularization-based methods were proposed. For example, Graph DropConnect Hasanzadeh et al. (2020) utilized random masks for channels. EGNNs Zhou et al. (2021) tried to limit the layer-wise Dirichlet energy. Inspired by ResNet He et al. (2015), residual connection-based methods are also proposed. For example, GCNII deployed residual connections to all layers of GNN.

**Knowledge graph embedding model** TransE Bordes et al. (2013) is the pioneering work proposed translation between node representations. After this work, various variants ( Wang et al. (2014), Lin et al. (2015)) have been proposed, to solve the limitation of translational methods such as one-to-many relations. However, models of this category have difficulties in learning the compositional and symmetry patterns. To overcome this limitation of translation-based models, rotation-based models were proposed. Distmult Yang et al. (2015) computes the Hadamard product between the head node and the edge, and the cosine similarity with the tail node is used as its scoring function. RotatE Sun et al. (2019) uses the representation of edge as the two-dimensional rotation in the complex space. QuatE Zhang et al. (2019) extends the idea of RotatE to the higher dimensional rotation.

**Self-supervised learning on HINs** The very early work of SSL in HINs was based on unsupervised learning with random walks. For example, PMNE Liu et al. (2017) proposed the method of maximizing the probabilities of the sampled neighbors. Metapath2vec Dong et al. (2017) was another pioneering work that deployed metapath-based random walk. These days, constrastive learning-based methods are gaining popularity. For example, inspired by DGI Veličković et al. (2018), HDGI Ren et al. (2020), DMGI Park et al. (2020) and HDMI Jing et al. (2021) proposed mutual information maximizing methods based on infomax theory. Various contrastive methods exhibited differences in generating augmented views from graphs. For example, MVGRL Hassani & Khasahmadi (2020) proposed contrast multi views on node level and graph level. HeCo Wang et al. (2021b) proposed Network schema view and metapath view for contrastive learning. In addition to the contrastive method, SHGP Yang et al. (2022) proposed SSL methods based on structural clustering.

## 6 CONCLUSION

In this study, we propose a novel SSL framework, BFTNet, a highly effective and robust SSL model for heterogeneous information networks. By separating the graph into features and topology, the model can optimize the learning of each one without sacrificing the performance of the other one, thereby improving the performance. Moreover, the model can maintain a stable training performance in noisy graphs by reflecting the relative importance of the graph topology and node features. BFTNet effectively generates integrated representation from node features and graph topology by utilizing the graph conversion scheme and the optimal modules. Extensive experiments demonstrated that our model outperforms baselines in all situations, with and without graph noises.

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
