# OpenReview forum: "Robust Self-supervised Learning in Heterogeneous Graph Based on Feature-Topology Balancing"
_ICLR.cc/2024/Conference — Submitted to ICLR 2024_

### Official Review · Reviewer_KdXi · 2023-10-22

**Soundness:** 4 excellent
**Presentation:** 4 excellent
**Contribution:** 3 good
**Rating:** 6
**Confidence:** 3

**Summary:**

This paper innovatively separates the learning of graph structure and node features, enhancing the model's capacity to distill rich information from both domains and optimize node embeddings. These advanced embeddings notably bolster performance in tasks like classification and clustering within IMDB, ACM, and MAG datasets. Crucially, the model demonstrates resilience when encountering unreliable information sources.

**Strengths:**

S1: The paper pioneers an approach that explicitly balances node feature and structure relevance in HIN learning under noisy conditions, addressing an authentic and ubiquitous challenge in real-world data scenarios. \
S2: Through empirical testing across diverse datasets, the proposed model not only registers substantial performance gains but also surpasses existing methods in robustness, validating its practical applicability. \
S3: The paper is commendable for its clarity, logical organization, and compelling presentation, making complex concepts accessible and the narrative persuasive. \
S4: The proposed model exhibits potential for scalability and adaptation to other types of networks or learning paradigms, making it a valuable reference point for future research endeavors. Its foundational concept is a launchpad for further explorations into HINs beyond the scope of the current study.

**Weaknesses:**

W1: The discussion on related works is sparse, especially considering that the balance between node features and graph structure isn't a novel concept in GNN research. A more detailed comparison with foundational works like [1] and [2] would better position and differentiate the paper's contributions within the field. \
W2: The paper introduces pivotal hyper-parameters alpha and beta without substantial exploration or practical guidelines for their optimization. This oversight diminishes the model's real-world utility. Incorporating a mechanism for their automatic adjustment, possibly based on mutual information or a related metric, could substantially augment the method's practicality. \
W3: The robustness assessment is limited, focusing on feature masking and edge-dropping. This narrow scope fails to fully stress-test the model's resilience. Expanding the range of adversarial challenges, including sophisticated strategies like MetaAttack [3], random feature corruption, or edge manipulation, would offer a more holistic robustness evaluation. \
W4: There is no substantial discussion on the model's computational demands. For real-world applications, particularly in larger, more complex networks, resource constraints are a vital consideration. The absence of this evaluation is a missed opportunity to understand the model's performance in resource-restricted environments.


[1] Ma, H., Liu, Z., Zhang, X., Zhang, L., & Jiang, H. (2021). Balancing topology structure and node attribute in evolutionary multi-objective community detection for attributed networks. Knowledge-Based Systems, 227, 107169. \
[2] Shi, M., Tang, Y., & Zhu, X. (2021). Topology and content co-alignment graph convolutional learning. IEEE Transactions on Neural Networks and Learning Systems, 33 (12), 7899-7907. \
[3] Daniel Zügner and Stephan Günnemann. 2019. Adversarial Attacks on Graph Neural Networks via Meta Learning. ICLR-2019

**Questions:**

Q1: Could you elucidate the distinctions between your method and pre-existing strategies for balancing node features and graph structure in HINs? \
Q2: What strategy would you recommend for the effective calibration of the introduced hyper-parameters, alpha and beta? \
Q3: Can the model maintain its performance integrity when exposed to an array of adversarial attacks beyond those discussed in the paper?

---

> ### Author Response · Authors · 2023-11-22
> **Response to Reviewer KdXi**
>
> Thanks for your comments. In the following, we respond to your concerns point by point.
> ***
> Point 1: Could you elucidate the distinctions between your method and pre-existing strategies for balancing node features and graph structure in HINs?
>
> In Reference [1], an evolutionary algorithm is applied to a homogeneous graph, and this methodology does not consider heterogeneous graphs, which have various types of nodes. Similarly, Reference [2] is also based on a homogeneous graph and uses a Graph Convolutional Networks (GCN) to learn node representations. This approach is vulnerable to the over-smoothing problem in large graphs. It can lead to incorrect representations being learned in graphs with incorrect topology, as the node features are updated based on the graph's topology.
> ***
> Point 2: What strategy would you recommend for the effective calibration of the introduced hyper-parameters, alpha and beta?
>
> As mentioned in the paper, alpha represents the balance between node features and topology. Therefore, if there is information about the quality of features and topology, this can be utilized to adjust alpha accordingly. Additionally, beta determines how much to align the distribution of features and topology. Thus, if the value of alpha is excessively biased, nearing 0 or 1, it is necessary to decrease the value of beta.
> ***
> Point 3: Can the model maintain its performance integrity when exposed to an array of adversarial attacks beyond those discussed in the paper?
>
> Thank you for your constructive feedback on aspects that were not fully considered in the paper. The need to consider adversarial attacks is noted, and we will aim to address this through further research in the future.

---

### Official Review · Reviewer_jyGu · 2023-10-23

**Soundness:** 2 fair
**Presentation:** 2 fair
**Contribution:** 2 fair
**Rating:** 3
**Confidence:** 4

**Summary:**

This work employs a knowledge graph embedding module focusing on global graph topology and a contrastive learning module dedicated to learning node features. Thanks to the novel structure that handles graph topology and node features separately, the proposed method assigns higher importance to one factor, thereby allowing it to effectively respond to skewed datasets in real-world situations. Moreover, the proposed method improves performance by designing the optimal module suited for learning the topology and features, without sacrificing the performance of one modality to reflect the characteristics of the other modality.

**Strengths:**

1. This work proposes to simultaneously capture the information in the graph topology and the information in the node features, which is reasonable.

2. Extensive comparison experiments and robustness experiments demonstrate the effectiveness of the proposed method.

**Weaknesses:**

1. The proposed method aims to separately learn the information in the graph topology and node features by the contrastive learning module and the knowledge graph embedding module, respectively. However, the knowledge graph embedding module seems also involves in the node features.

2. The novelty of the proposed method is relatively low. The contrastive learning module is the same as the InfoNCE loss in previous methods. Moreover, directly align the topology-based representations and feature representations seems not very reasonable.

3. The proposed method lacks theoretical analysis. For example. Why the proposed method can outperform the previous methods, are there any theoretical supports?

4. Does the proposed framework in this paper really enable topological maps and node features to complement each other, and are there any case studies to validate this?

5. The writing of the proposed method needs more improvements, and the manuscripts needs more proofreading. For example,
in page 6, "six Self-supervised heterogeneous models" should be "six self-supervised heterogeneous models".
It's best not to take a direct screenshot of the framework figure.

6. Too few comparison methods, especially lack of 2023 updates.

7. This work was not provided in code, making it difficult to measure its reproducibility.

Based on the above comments, I think that the work does not meet the ICLR thresholds.

**Questions:**

See above.

---

> ### Author Response · Authors · 2023-11-22
> **Response to Reviewer jyGu**
>
> We sincerely appreciate your comments. In the following, we present our response to address your concerns.
> ***
> Point 1. The knowledge graph embedding module also seems involved in the node features.
> I would like to respectfully point out that the concern mentioned contains a factual error. In the case of the knowledge graph module, all node features are initialized randomly, and the information from node features is not used throughout the pre-training process. The only aspect in which node features influence the training of the knowledge graph module is through the MSE loss. This allows for effective adjustment of weights for each component of the graph.
> ***
> Point 2. Does the proposed framework in this paper really enable topological maps and node features to complement each other, and are there any case studies to validate this?
> Thank you for your insightful feedback. The authors believe that Table 4, which presents the results of the ablation study, demonstrates that topology and features have complementary characteristics. In fact, when either the feature or topology is excluded, the model shows a significant decline in performance. By utilizing both, the shortcomings of each modality can be compensated for, allowing for improved representation learning.
> ***
> Thank you for the detailed review. In future research, we will include additional comparison methods and conduct theoretical analyses to enhance the completeness of our research. Thank you.

---

### Official Review · Reviewer_69NY · 2023-10-26

**Soundness:** 3 good
**Presentation:** 3 good
**Contribution:** 3 good
**Rating:** 3
**Confidence:** 5

**Summary:**

This paper introduces a framework called BFTNet for robust self-supervised learning in heterogeneous graphs. The framework separates the graph topology and node features, allowing it to effectively handle skewed datasets and improve performance without compromising either modality. The proposed approach is validated through extensive experiments on real-world benchmark datasets, demonstrating its robustness even in noisy environments.

**Strengths:**

1. This paper highlights a novel graph conversion scheme as well as representation fusion methods employed within BFTNet that ensure effective learning integration between topology and features.
2. This paper addresses key challenges in self-supervised learning on complex heterogeneous graphs.

**Weaknesses:**

1. This paper does not discuss potential scalability issues or computational complexity associated with implementing BFTNet on large-scale heterogeneous graphs.
2. Limited discussion on the interpretability and explainability of the BFTNet framework. While it is mentioned that BFTNet separates graph topology and node features, allowing for improved performance, there is no in-depth analysis or explanation provided regarding how this separation enhances interpretability or facilitates understanding of the underlying relationships within heterogeneous graphs. This lack of clarity may hinder researchers' ability to fully comprehend and utilize the insights gained from using BFTNet in their own studies.
3. In Experiments, It appears that this paper achieves dominant performance over baselines. However it does not directly point out how to get the results. Is it the maximum value or the average value? Besides, although it is stated that source code for implementing BFTNet will be made available in future versions, at present it remains unavailable, which limits reproducibility.
4. This paper uses masking node features and dropping edges to verify robustness. However, these two ways are too simple. More test, such as Meta attack [1], is needed.

**Questions:**

see above

---

> ### Author Response · Authors · 2023-11-22
> **Response to Reviewer 69NY**
>
> Thank you very much for your sincere and detailed review of our research. Firstly, I apologize for any confusion caused to the reviewers due to our failure to clearly articulate that the reported performance is average after repeated experiments. We regret not having written a clearer and more refined paper. Furthermore, we will consider the issues of scalability and explainability that you have highlighted and aim to develop more comprehensive research in the future. Lastly, we will conduct experiments in a variety of graph perturbation settings, including adversarial attacks, to thoroughly assess the performance of our model. Once again, I would like to express my gratitude to the reviewers for helping our research move in a better direction.

---

### Meta-Review · Area_Chair_PUtY · 2023-12-10

**Metareview:**

I recommend to reject this paper.

  The paper presents BFTNet, a framework for robust self-supervised learning in diverse graphs. It effectively handles skewed data by separating graph topology and node features. The authors also validated the proposed method through extensive real-world experiments.

  As pointed out by reviewers, although the problem focused in this paper is practical and important, but the paper lacks discussion on scalability and interpretability, while methodological clarity and robustness testing using more comprehensive approaches are needed. I would encourage the authors to take the reviewers' feedback to further revise the paper As a result, I recommend to reject this paper.

**Justification For Why Not Higher Score:**

N/A.

**Justification For Why Not Lower Score:**

N/A.

---

### Decision · Program_Chairs · 2024-01-16

Reject